# *Terminalia petiolaris* A.Cunn ex Benth. Extracts Have Antibacterial Activity and Potentiate Conventional Antibiotics against β-Lactam-Drug-Resistant Bacteria

**DOI:** 10.3390/antibiotics12111643

**Published:** 2023-11-20

**Authors:** Muhammad Jawad Zai, Matthew James Cheesman, Ian Edwin Cock

**Affiliations:** 1Centre for Planetary Health and Food Security, Griffith University, Brisbane, QLD 4111, Australia; muhammadjawad.zai@griffithuni.edu.au (M.J.Z.); i.cock@griffith.edu.au (I.E.C.); 2School of Environment and Science, Griffith University, Brisbane, QLD 4111, Australia; 3School of Pharmacy and Medical Sciences, Griffith University, Southport, QLD 4222, Australia

**Keywords:** *Terminalia petiolaris*, antibiotic-resistant, ESBL, MRSA, flavonoids

## Abstract

*Terminalia petiolaris* A. Cunn. Ex Benth. (genus: *Terminalia*, family: Combretaceae) is native to Australia. *Terminalia* spp. have traditionally been used to treat various ailments, including bacterial infections. Solvents of varying polarity were used to extract compounds from leaves of this species, and the extracts were tested against a panel of bacteria, including antibiotic-resistant strains. The methanolic and water extracts showed substantial inhibitory activity against several bacteria, including antibiotic-resistant strains in both disc diffusion and liquid dilution assays. Combining these extracts with selected conventional antibiotics enhanced the inhibition of bacterial growth for some combinations, while others showed no significant interaction. In total, two synergistic, twenty-five additive, twenty-three non-interactive and one antagonistic interaction were observed. The methanolic and ethyl acetate plant extracts were found to be non-toxic in *Artemia franciscana* nauplii toxicity assays. A liquid chromatography–mass spectrometry metabolomics analysis identified several flavonoid compounds, including miquelianin, trifolin and orientin, which might contribute to the observed activities. The potential modes of these active extracts are further discussed in this study.

## 1. Introduction

The emergence of antibiotic-resistant bacterial pathogens poses an increased risk to global human health. According to the World Health Organization (WHO), it is estimated that by 2050, the annual death toll due to antimicrobial resistance (AMR) will surpass the number of deaths caused by cancer, signifying a critical threat to public health worldwide [1]. Despite the ongoing SARS-CoV-2 pandemic, the WHO has affirmed that antimicrobial resistance (AMR) continues to be the foremost global public health threat confronting humanity [2]. Antimicrobial resistance (AMR) does not discriminate based on geographical location or income levels. Untreatable common infections are surfacing at alarming rates, leading to increased patient morbidity and mortality worldwide [3,4].

Children are particularly susceptible to multidrug-resistant (MDR) pathogens and have lower inherent immunity. Pathogenic diseases account for ~40% of childhood mortalities and morbidity deaths, particularly in developing countries [5]. Indeed, antibiotic-resistant neonatal infections affect approximately 60,000 newborn babies in India alone [6]. The Gram-positive bacteria *Staphylococcus aureus* and the Gram-negative species *Escherichia coli* and *Klebsiella* spp. are of particular concern. The prevalence of pneumonia caused by *S. aureus* in children has increased substantially, while the increasing incidence of methicillin-resistant *S. aureus* (MRSA) has increased hospitalizations and fatalities [7,8]. Similarly, *Klebsiella pneumoniae* is responsible for a wide array of infections, including various types of respiratory, urinary tract and bloodstream infections [9], while *E. coli* is a significant cause of diarrhea, particularly in children [10]. Clinical isolates expressing extended-spectrum β-lactamases (ESBLs) have been identified in both *K. pneumoniae* and *E. coli*, indicating the alarming spread of antibiotic resistance in these pathogens [11,12]. Notably, ESBL-expressing strains of *K. pneumoniae* and *E. coli* exhibit substantially elevated levels of resistance [13,14]. Consequently, treatment options for infections caused by these organisms are severely limited, posing a significant challenge to healthcare professionals [15].

The swift rise in antimicrobial resistance (AMR) has significantly diminished the efficacy of numerous classes of antibiotics. Consequently, there has been a surge in investigations aimed at exploring new treatment approaches for infections caused by both antibiotic-susceptible and multidrug-resistant (MDR) bacterial species. Traditional plant-based medicines have garnered substantial interest in recent years as potential targets for new drug development. This interest stems from mounting evidence of their antimicrobial properties and their ability to augment the effectiveness of conventional antibiotics [16]. The genus *Terminalia* comprises approximately 200–250 species, many of which have a rich history of traditional usage for various ailments [17]. Notably, many *Terminalia* species exhibit high antioxidant capacities. *Terminalia ferdinandiana* Exell has been reported to possess the highest antioxidant activity of any plant globally. Indeed, the levels of ascorbic acids in this species surpass those of blueberry by more than 900 times [18]. Although many of the compounds in *Terminalia* species remain unidentified, several flavonols, flavonoids and benzoic acids have been identified in some *Terminalia* spp. [19]. *Terminalia petiolaris* Benth is an endemic Australian *Terminalia* spp. that has previously been reported to inhibit the growth of a limited panel of bacterial pathogens [20]. Notably, that study reported that *T. petiolaris* inhibits some strains of *K. pneumoniae*. This species is rich in phenolic compounds, including gallic acid, chebulic acid, flavonoids, and terpenoids.

There is a lack of studies examining extracts or compounds derived from *T. petiolaris* against MRSA or other antibiotic-resistant strains of bacteria, highlighting the need for further research in this area. The objective of this study was to evaluate *T. petiolaris* extracts against a panel of antibiotic-sensitive bacterial species, as well as corresponding antibiotic-resistant variants. Specifically, this study screened *S. aureus* and a methicillin-resistant strain (MRSA), *E. coli* and ESBL *E. coli*, and *K. pneumoniae* and ESBL *K. pneumoniae*. These bacteria contribute substantially to bacterial infection-induced fatalities and often display high levels of antibiotic-resistance. Liquid chromatography–mass spectrometry (LC-MS) analysis of the *T. petiolaris* extracts was conducted to identify and highlight notable compounds within the extracts. The toxicities of the plant extracts were determined using an *Artemia franciscana* (brine shrimp) nauplii toxicity assay.

## 2. Results

### 2.1. Antibacterial Susceptibility Studies

The extraction of dried and ground *T. petiolaris* leaves with various solvents, subsequent drying and resuspension in 10 mL of 1% DMSO resulted in extracts with concentrations of 14, 7 and 2 mg/mL for the methanolic, aqueous and ethyl acetate extracts, respectively. In this study, the susceptibility and resistance of the bacterial strains were evaluated using selected antibiotics from diverse classes. The activities were assessed both on agar plates (measured as zones of inhibition (ZOIs)) and in broth microdilution assays and expressed as minimum inhibitory concentrations (MICs). Notably, the results obtained from the agar (Figure 1) and broth methods (Table 1) for each antibiotic were generally consistent. The *T. petiolaris* methanol (TPM) and water (TPW) extracts inhibited the growth of all six bacterial strains used in this study in both the disc diffusion (Figure 1) and liquid dilution assays (Table 1). Interestingly, the ethyl acetate (TPE) extract only inhibited the growth of the *E. coli*, ESBL *E. coli* and MRSA strains in the disc diffusion assay. The ethyl acetate extract was also effective in the broth assays against all the bacteria tested, with MIC values of 250 µg/mL recorded against all strains. Noteworthy antibacterial activity was noted for the methanolic extract against all bacterial strains. The methanolic extract was particularly effective against *E. coli* (MIC = 109.4 µg/mL), although noteworthy activity was also noted against the other bacterial strains, with an MIC of 437.5 µg/mL (Table 1). Interestingly, the aqueous extract had similar efficacy against both the ESBL *K. pneumoniae* (MIC = 212.5 µg/mL) and the corresponding sensitive *K. pneumoniae* strain (MIC = 218.75 µg/mL). This finding suggests that TPW extract may be more effective in treating infections caused by resistant strains of *K. pneumoniae* compared to sensitive strains of *K. pneumoniae*. However, this effect was not observed for *S. aureus* and MRSA, where the MIC values were similar.

### 2.2. Determination of Fractional Inhibitory Concentration

A diverse array of interactions was observed when the *T. petiolaris* leaf extracts were combined with conventional antibiotics against both antibiotic-sensitive and resistant strains of *E. coli*, *S. aureus* and *K. pneumoniae* (Table 2). In the combinational studies, two combinations (TPW with tetracycline and TPE with erythromycin) exhibited synergistic effects, while twenty-five combinations were additive, twenty-three were non-interactive and one combination (TPW with gentamicin) displayed an antagonistic effect.

### 2.3. Synergistic Interaction of Extract–Antibiotic at Different Ratios

Two extract–antibiotic combinations yielded synergy, and these included TPW in combination with tetracycline and TPE in combination with erythromycin (Table 2). Therefore, various different ratios of these combinations were evaluated and plotted as isobolograms to identify the ideal ratios at which synergy occurs. Only those ratios that yielded synergistic and additive interactions were included in the isobolograms (Figure 2). Notably, the TPW extract and tetracycline combination exhibited synergistic inhibition against *S. aureus* only in ratios containing 20–50% extract. In contrast, ratios containing 10% extract, as well as those containing 60–80% extract, in the combinations produced additive effects. While these combination ratios would provide additional benefits compared to using either component separately against *S. aureus*, they would not be as effective as the synergistic ratios. The combination containing 90% TPW and 10% tetracycline resulted in indifferent effects, indicating that they did not offer any advantages over using the individual components alone. Notably, the only ratio of the TPE and erythromycin combination that exhibited a synergistic inhibitory effect against *K. pneumoniae* was 60% extract:40% erythromycin. Importantly, all the other ratios resulted in indifferent effects, signifying that they did not provide any additional advantages over using the individual components separately.

### 2.4. Identification of Compounds in T. petiolaris Methanol (TPM) and Aqueous (TPW Extracts)

As the methanolic and aqueous extracts had the greatest antibacterial activity in both the disc diffusion and liquid dilution assays, they were deemed the most promising extracts for phytochemical identification studies. The optimized HPLC-MS parameters described in Section 4.9 were developed and used to examine the metabolomic fingerprint of those extracts, focusing on the flavonoid components. The resultant total compound chromatogram in the positive ionisation mode for TPM and TPW are presented in Figure 3a and Figure 3b, respectively. Notably, most of the extract compounds eluted during the isocratic stage of 95% acetonitrile, indicating that the extracts contained mainly polar compounds. Compound Discoverer software (version 3.3; Thermo Scientific, Waltham, MA, USA) was used to identify the individual compounds in the extracts. Compound Discoverer was set to screen the detected mass signals against Natural Product Unknown ID with Stats Online and Local Database Searches. The total ion chromatograms of the individual plant extracts and blank controls were compared using the Compound Discover software (version 3.3). Empirical formulas were determined according to the principles of the isotope abundance ratio and nitrogen rule combined with Xcalibur software (4.5; Thermo Scientific, Waltham, USA). During the measurements, the deviation between the m/z values in the primary mass spectrum and the theoretical m/z was found to be less than 5 ppm, affirming the accuracy of the instrument’s measurement results. Additionally, the fragment information obtained from both primary and secondary mass spectra was compared with the ChemSpider, m/zCloud and Masslist databases, and only those compounds that fully matched any of the databases (and were not present in the blank controls) were selected. Additionally, an extensive literature search was also performed to enhance the confidence in the results. A range of different compounds were identified in the TPM and TPW extracts, of which fourteen and three were flavonoids, respectively (Table 3).

### 2.5. Toxicity Quantification

To assess toxicity, the extracts were evaluated against an *Artemia franciscana* nauplii toxicity assay within a concentration range of 125 to 1000 μg/mL. The average percentage of mortality from three repeated experiments was used to calculate the LC_50_ values. Extracts causing less than 50% mortality at a specific concentration were considered non-toxic at that dose. For extracts inducing more than 50% mortality, further dilutions were made and tested across various concentrations until a level with less than 50% mortality was identified. Notably, both the methanolic and ethyl acetate extracts induced <50% mortality at 1000 μg/mL and therefore were deemed to be nontoxic. In contrast, the aqueous extract induced >50% mortality at 1000 μg/mL and was therefore further diluted and tested. The LC_50_ of the TPW extract was determined to be 250 μg/mL, and therefore the TPW extract was deemed to be toxic. In contrast, 100% mortality was induced by 4 mg/mL of potassium dichromate (the positive control), and 0% mortality was induced by seawater (the negative control).

## 3. Discussion

This study determined that both the methanolic and aqueous extracts of *T. petiolaris* inhibited the growth of all six bacterial strains tested in the disc diffusion and liquid dilution assays. The methanolic extracts demonstrated the greatest potency in inhibiting bacterial growth. In contrast, the ethyl acetate extract inhibited the activity of all strains in the liquid dilution assay but few strains in the disc diffusion assay. These differences may be linked to the varying yields and types of phytochemicals extracted from *T. petiolaris* using different solvents. Typically, solvents with lower polarity extract fewer phytochemicals from plants compared to polar solvents [21]. Methanol and water each extracted a wide array of compounds, including tannins and flavonoids, while ethyl acetate, which is less polar, extracted substantially fewer phytochemicals [22]. These differences in phytochemical composition may explain the varying levels of bacterial growth inhibition observed in both the liquid dilution and disc diffusion assays. Phytochemicals that are less polar or larger in size may not diffuse through the solid agar, hence impacting the antimicrobial effectiveness of the extracts [23]. Additionally, volatile compounds tend to evaporate from agar gel, reducing their concentration in the extract and, therefore, their efficacy [23]. Disc diffusion assays are also influenced by compound solubility; polar compounds diffuse faster than less soluble ones, which remain concentrated around the disc, potentially leading to underestimated MIC values [24]. The liquid dilution assay is considered to be more sensitive than the disc diffusion assay because it is not affected by the size and polarity of compounds.

Notably, the *T. petiolaris* extracts exhibited comparable inhibitory effects on antibiotic-resistant bacterial species compared to their susceptible counterparts of the same bacterial species. This suggests that the active compounds in the *T. petiolaris* extracts are relatively unaffected by the resistance mechanisms found in the ESBL and MRSA strains. Thus, the *T. petiolaris* extract compounds may exert their effects via different mechanisms from the antibiotics that these strains have developed resistance to, including β-lactams. Alternatively, these extracts may contain components that can counteract the antibiotic-resistance mechanisms within these bacterial strains. This discovery is promising because the antibiotic-resistant bacteria tested in this study displayed substantially lower susceptibilities to multiple antibiotics from different classes, including β-lactams, macrolides, sulfonamides, fluoroquinolones, aminoglycosides and tetracyclines, compared to their susceptible counterparts. Moreover, the *T. petiolaris* extracts demonstrated effectiveness against both Gram-positive and Gram-negative bacteria, indicating their potential as broad-spectrum antibiotics. These extracts should be screened against a wider panel of bacterial pathogens to obtain a more complete picture of their antibiotic potential.

Recent studies have demonstrated the potential of combining active plant extracts with conventional antibiotics to enhance their antibacterial effects against various bacterial strains [16]. In our study, two synergistic, twenty-five additive, twenty-three non-interactive and one antagonistic interaction were noted. Synergistic interactions greatly improve antibacterial effectiveness in comparison to additive interactions, which also enhance antibiotic efficacy, although not as significantly as synergistic interactions do. Synergistic interactions were observed when TPW was combined with tetracycline and tested against *S. aureus* and the TPE extract was combined with erythromycin against *K. pneumoniae*. Additionally, our study confirmed the non-toxic nature of the TPM and TPE *T. petiolaris* extracts in *Artemia* nauplii toxicity assays, further supporting their safe use in combination therapies. In contrast, while non-interactive combinations do not enhance antibacterial activity compared to either component individually, they also do not decrease the effectiveness of the individual components, making them safe for concurrent use. However, antagonistic interactions decrease the overall activity of extract–antibiotic combinations and should be avoided.

Several previous studies used LC-MS metabolomics analyses to profile the phytochemical composition of other *Terminalia* spp. for the presence of specific compounds and to quantify their relative abundances [25]. Notably, several compound classes were detected in the leaf extracts of multiple *Terminalia* spp. In particular, low-molecular-mass tannins, including ellagic acid, gallic acid and chebulic acid, were present in relative abundance in several species. Additionally, higher-molecular-mass tannins, including chebulagic acid, castalagin, corilagin and chebulinic acid, have also previously been reported in relative abundance in the leaf extracts of multiple *Terminalia* spp. leaf extracts. These compounds were also identified in relative abundance in the *T. petiolaris* methanol and water extracts examined in our study. However, this study primarily concentrated on the analysis of flavonoids. Flavonoids are widely recognized for their antibacterial properties against various pathogenic microorganisms [26]. Given the rising incidence of untreatable infections caused by antibiotic-resistant bacteria, flavonoids have garnered significant attention due to their potential as alternatives to antibiotics and their ability to overcome some bacterial resistance mechanisms.

Some studies have assessed the inhibitory effects of plant extracts rich in flavonoids as well as pure flavonoid compounds against several pathogenic bacteria. The antibacterial activities of flavones have been attributed to multiple mechanisms. One such mechanism involves the formation of complexes between flavones and cell wall components, thereby disrupting further adhesion and inhibiting microbial growth [27]. For example, amentoflavone and gancaonin isolated from *Dorstenia barteri* Bureau and *Dorstenia angusticornis* Engl. inhibited the growth of *Bacillus cereus* (MIC = 2.4 and 3 µg/mL, respectively) via altering the cell wall [28,29]. Similarly, the flavone baicalein is an effective bactericide, and when combined with cefotaxime, it synergizes the effects of an antibiotic [30]. Additionally, baicalein reduces the *Pseudomonas aeruginosa* secretion of the pro-inflammatory cytokines IL-1β, IL-6 and TNFα, which are important inflammatory mediators following *P. aeruginosa* infection [31]. Similarly, other studies have reported that baicalein, when present at concentrations of 32 and 64 μg/mL, suppresses the quorum-sensing system regulators agrA, RNAIII and sarA and downregulates the gene expression of intercellular adhesin in *S. aureus* cells. This is noteworthy as adhesion is required by *S. aureus* cells to produce biofilms, and its inhibition may therefore increase the sensitivity of this bacterium to antibiotics.

Similarly, the flavonoids quercetin, kaempferol, and luteolin, as well as their derivatives, have been reported to have potent antibacterial activities. Notably, derivatives of these flavonoids are present in the *T. petiolaris* extracts, namely miquelianin (quercetin 3-O-glucuronide) (Figure 4A), trifolin (kaempferol 3-O-β-D- galactopyranoside) (Figure 4B) and orientin (8-C glucoside of luteolin) (Figure 4C), as indicated by the LC-MS analysis (Table 3). Quercetin and its derivatives have been particularly well studied and reported to have significant antibacterial activity against a panel of bacteria, including *S. aureus*, MRSA and *Staphylococcus epidermis*. Additionally, in vitro studies investigating quercetin’s effect on several oral microbes reported potent activity against *Porphyromonas gingivalis,* with an MIC value of 0.0125 µg/mL [32]. Another study evaluated quercetin’s antibacterial effects on amoxicillin-resistant *S. epidermis* and reported that when quercetin was combined with amoxicillin, synergistic activity was observed, effectively reversing bacterial resistance to this conventional antibiotic [33]. Studies have also reported the antibacterial activity of kaempferol against *E. coli*, with an MIC value of 25 µg/mL, which was substantially less than that of quercetin and luteolin (MIC = 35.76 and 67.25 µg/mL, respectively) [34]. The difference in the MICs among these flavonoids is due to their structural differences. Quercetin has a hydroxyl group at position 3 in the C ring, while luteolin lacks this group. This indicated that the hydroxyl group at position 3 is important to the antibacterial activity of quercetin [35]. In this study, the structure analogues of quercetin, kaempferol and luteolin were identified in the methanolic and aqueous extract of *T. petiolaris*. These analogues may exhibit similar or better antibacterial activity than their parent flavonoids, although this remains to be verified.

The flavonoids identified in the *T. petiolaris* extracts in our study may have inherent antibacterial properties and/or enhance the activities of other phytochemicals in *T. petiolaris* and of some conventional antibiotics. However, further studies are required to determine their effects (if any). Future research should also explore these compounds as potential scaffolds for new antibiotics or as potentiators for existing antibiotics. Notably, the *T. petiolaris* extracts were effective against both antibiotic-sensitive and resistant bacteria in our study, which is a desirable trait for newly developed drugs or formulations. To advance this approach, the flavonoids identified herein need comprehensive characterization. Evaluations of their inherent antibacterial properties, both alone and in combination with other phytoconstituents (such as tannins, polyphenols, cardiac glycoside, etc.), are required to determine their antibacterial potency and their specific mechanisms. Additionally, certain compounds identified in the LC-MS metabolomics profiling analysis conducted in this study remain unidentified. It is possible that these compounds may also contribute to the antibacterial activity of the examined extracts, either as antibacterial agents or enhancers. Further phytochemical investigation studies are required to identify these compounds and to fully explore their potential activities.

The methanol and ethyl acetate extracts screened in this study were nontoxic in the *Artemia* nauplli bioassay, indicating that they are safe for antibiotic use. However, further testing in a panel of human cell lines is required before these extracts are used medicinally. Of concern, the aqueous extract displayed toxicity in this assay, and therefore it should only be used with caution. However, the fractionation of that extract may be able to separate antibacterial compounds from toxic compounds. Further studies are required to explore this possibility.

## 4. Materials and Methods

### 4.1. Materials

All solvents utilized in this study were of analytical grade and were sourced from Ajax Fine-Chemicals Ltd., Taren Point, Australia. Mueller–Hinton agar and broth were procured from Oxoid Ltd., Thebarton, Australia. Unless otherwise specified, all additional chemicals were obtained from Sigma Aldrich, Clayton, Australia.

### 4.2. Plant Collection and Extraction

Jacinta Monck from Kimberley Wild Gubinge, Australia, supplied the *T. petiolaris* leaves used in these studies. The plant material was supplied as fresh leaves, and a voucher specimen was stored in the School of Environment and Science, Griffith University (TP_WAJM21). The leaves were dried using a Sunbeam food dehydrator and then ground into a fine powder. One-gram portions of the leaf powder were weighed into individual tubes. Methanol, water and ethyl acetate (50 mL each) were added separately and mixed thoroughly. The plant material was extracted at room temperature by maceration for 24 h. The particulate material was then removed by vacuum filtering the extracts through Whatman No. 54 filter paper. The extracts were dried under vacuum conditions, and the mass of the extracted material was weighed. A volume of 100 μL DMSO was added to the dried extract, and the volume was made up to 10 mL with sterile deionized water and allowed to sit at room temperature for 24 h. Multiple 30 s sonication cycles were used to assist with extract solubilization, and the particulate matter was removed by syringe filtration with 0.22 μm syringe-driven filters (Millipore Australia Pty Ltd., North Ryde, Australia). The extracts were aliquoted and stored at 4 °C until use.

### 4.3. Antibacterial Studies

A modified disc diffusion assay was used for initial antibiotic susceptibility screening studies. A checkerboard liquid micro-dilution assay was used to quantify the MIC of each extract, as well as for the control antibiotics and combinations. This method is generally more sensitive than the disc diffusion method, as it is perhaps the most widely used method to determine MIC values, so it allows for comparisons between studies and between extracts.

### 4.4. Growth of Bacterial Cultures

Meuller–Hinton agar and broth powders were obtained from Oxoid Ltd. (Thebarton, Australia) and prepared following the manufacturer’s instructions. Each bacterial stock culture was initially streaked onto an agar plate to confirm the purity of the culture and to obtain pure colonies for testing. The agar plates were incubated at 37 °C for 24 h, and then a single pure colony of the bacteria was inoculated into fresh Meuller–Hinton broth and incubated at 37 °C until log growth was achieved. This culture was used to prepare a stock culture by inoculating 100 μL of the bacterial culture into a fresh nutrient medium and incubating at 37 °C for 24 h.

### 4.5. Disc Diffusion Assay Screening

Modified Kirby–Bauer disc diffusion assays were used as initial screens for antibiotic susceptibility [22]. Briefly, Mueller–Hinton agar plates were individually spread with a 100 µL volume of each bacterial suspension and incubated for two hours at room temperature. Filter paper discs were infused with 10 µL of the extracts and placed onto the agar surface. Standard discs containing ampicillin (2 μg) or erythromycin (10 μg) (Oxoid Ltd., Thebarton, Australia) and discs infused with 10 μL of sterile deionized water were included on agar plates as positive and negative controls, respectively, following incubation at 37 °C for 24 h. Each assay was conducted in triplicate (*n* = 3), and the results were reported as mean values (±SEM).

### 4.6. Liquid Microdilution MIC Assay

The MIC values were determined using liquid microdilution assays. These assays were used to quantify the MIC values of the extracts or antibiotics as monotherapies, as well as evaluate the FIC values of combinations consisting of extracts and control antibiotics. Individual 0.5 MacFarland standard bacterial cultures were prepared from the stock cultures and further diluted to 1 part stock culture to 100 parts fresh Mueller–Hinton broth, resulting in approximately 1 × 10^6^ colony-forming units/mL.

A volume of 100 μL of sterile nutrient broth was added to all wells of a 96-well plate. Then, 100 μL of the test solution (comprising either an antibiotic, extract or extract–antibiotic combination in a 1:1 ratio) was added and mixed. Serial two-fold dilutions were performed down each column of the plate, resulting in a final volume of 100 μL per well. The plate included sterile controls (without bacteria), negative controls (containing sterile water), positive controls (with control antibiotics) and growth controls (without any test solution). Additionally, MacFarland standard cultures were streaked onto Mueller–Hinton agar plates on the same day as the MIC quantification experiments to ensure the purity of the culture and the absence of microbial contamination. Subsequently, 100 μL volume of the bacterial culture was added to each well of the plate, which was then incubated at 37 °C for 24 h. Each assay was conducted twice in duplicate (*n* = 4), and the mean of the MIC values is reported in the study.

*p*-Iodonitrotetrazolium violet (INT) was purchased from Sigma (Australia) and dissolved in sterile deionized water to prepare a 0.4 mg/mL INT solution. After the initial 24 h incubation, 40 μL of the INT solution was added to all wells on the plates, followed by a further 6 h of incubation at room temperature. The MIC was visually determined as the lowest dose at which the development of colour was inhibited.

### 4.7. Determination of Combinational Effects

Combinational studies were used to assess the interactions between the extracts and the conventional antibiotics. For these tests, the extracts and antibiotics were combined in 1:1 ratios. The assays were performed as above, and the following formulas were subsequently used to quantify fractional inhibitory concentration (FIC) values:FIC (E) = (MIC of plant extract in combination with antibiotic)/(MIC of plant extract independently) 
FIC (A) = (MIC of antibiotic in combination with plant extract)/(MIC of antibiotic independently)
ΣFIC = FIC (E) + FIC (A)

ΣFIC values ≤ 0.5 were classified as synergistic; 0.5–≤1.0 were classified as additive; >1.0–≤4.0 were classified as non-interactive/indifferent; and ΣFIC values > 4.0 were classified as antagonistic.

Combinations that showed synergy were further evaluated to determine their optimum concentrations. This assessment was based on a colourimetric analysis that allowed the MIC to be determined as the lowest dose inhibiting the development of colour. Each plate included a negative control (water), positive control (antibiotic), sterile control (without inoculum); and growth control (without any test solution). Initial screenings were conducted in triplicate (*n* = 3).

### 4.8. Determination of Optimal Ratios by Isobologram Analysis

To investigate combinations that exhibit synergy, their optimal ratios were determined. The same procedure as described in the liquid dilution assay was followed, with variations in the ratios of antibiotic–extract combinations tested. The ratio of conventional antibiotics to plant extracts was sequentially reduced by 10 μL from an initial volume of 100 μL, ultimately reaching 0 μL, creating combinations ranging from 100% to 0% extract with 10% decreasing increments. Simultaneously, the amount of antibiotic was increased by 10 μL, starting from 0 μL and reaching 100 μL.

All assays were conducted in duplicates. The acquired data were used to calculate FIC values, and an isobologram analysis was employed to determine the ratios at which a synergistic interaction occurred between plant extract A and antibiotic B. Interactions resulting in a ΣFIC value of ≤0.5 were considered synergistic, while ΣFIC values between 0.5 and 1.0 were considered additive. Interactions with ΣFIC values greater than 1.0 but not exceeding 4.0 were classified as non-interactive, and interactions with ΣFIC values greater than 4.0 were deemed antagonistic.

### 4.9. Non-Targeted Headspace LC-MS Conditions for Quantitative Analysis

Non-targeted headspace metabolic profiling was used to identify extract components. LC-MS was conducted using a Vanquish Ultra High-Performance Liquid Chromatography (UHPLC) system (Thermo Fisher Scientific). Separation was achieved using a Waters Acquity UPLC BEH amide column (2.1 mm × 100 mm, 1.7 μm), connected to an Orbitrap Exploris 120 mass spectrometer (Orbitrap MS, Thermo Fisher). The UHPLC system utilized a quaternary pump at a flow rate of 0.6 mL/min, using the following mobile phases: (A) 0.1% *v/v* formic acid in ultrapure water and (B) acetonitrile (MeCN) containing 0.1% *v/v* formic acid. The system was programmed to run the following gradient: 5% to 95% B for 16.50 min and 2 min of isocratic elution at 95% B to flush the column. The column was subsequently reequilibrated by decreasing the mobile phases to 5% B and running isocratically at 5% B for 2 min.

The mass spectra of the eluted compounds were analyzed using an Orbitrap Exploris 120 mass spectrometer in information-dependent acquisition (IDA) mode. The system was controlled by Xcalibur acquisition software (Thermo Fisher). The electrospray ionization (ESI) utilized the following parameters: a 63.72 psi sheath gas pressure; a 10.39 psi auxiliary gas pressure and a 320 °C capillary temperature. Full MS resolution was achieved at 60,000, while MS/MS resolution was achieved at 15,000. The collision energy in NCE mode was set at 10/30/60/70, with a spray voltage (positive) of 3.4 kV. The data were analyzed using Compound DiscovererTM 3.2 software with a Fragment Ion Search (FISh) function. The putative compound identification (where possible) and molecular annotation used the Global Natural Product Social Molecular Networking (GNPS), mzCloud, mzVault, CyanoMetDB and Chemspider databases and were compared to published data.

### 4.10. Toxicity Studies

*Artemia franciscana* nauplii lethality assays (ALA) were used to evaluate the toxicity of the extracts. Volumes of 400 µL of artificial seawater (Red Sea Pharm Ltd., Pituach, Israel) containing approximately 50 *A. franciscana* nauplii were added into individual wells of a 48-well plate, and 400 µL of individual plant extracts, reference toxin or the seawater negative control were added to the wells. The plates were incubated at 25 ± 1 °C for 24 h, and the % of dead nauplii in each well was determined and used to calculate the concentration that induced 50% mortality (LC_50_). All test plates included wells treated with 400 µL of potassium dichromate (test concentration of 1 mg/mL) as a positive control and wells treated with 400 µL of artificial seawater (Red Sea Pharm Ltd., Pituach, Israel) as a negative control. All ALA tests and control evaluations were conducted in triplicate (*n* = 3).

## 5. Conclusions

The urgent need for new therapies to combat bacterial infections has led to increased exploration of natural products as potential sources of antibiotics. Our research indicates that *T. petiolaris* extracts effectively inhibit the growth of multidrug-resistant bacteria to a similar extent as for sensitive strains. This suggests that certain components within the plant extracts may possess unique and previously undiscovered antibacterial mechanisms. Additionally, some of the compounds identified in this study could be partially responsible for the observed activities. Future studies should focus on further investigating these compounds as potential antibacterial agents. Moreover, exploring their potential to enhance the effectiveness of conventional antibiotics or other phytochemicals may also be a key area of research.

## Figures and Tables

**Figure 1 antibiotics-12-01643-f001:**
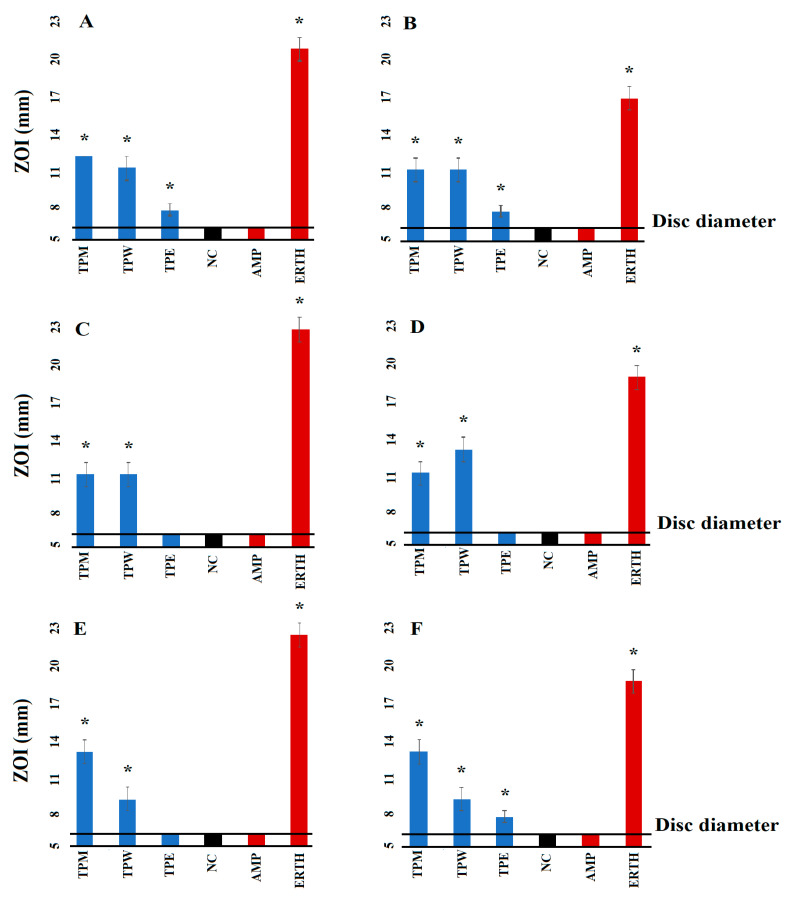
Antimicrobial activity of the plant extracts against (**A**) *E. coli*, (**B**) ESBL *E. coli*, (**C**) *S. aureus*, (**D**) MRSA, (**E**) *K. pneumoniae* and (**F**) ESBL *K. pneumoniae*. TPM = *Terminalia petiolaris* methanol; TPW = *Terminalia petiolaris* water; TPE = *Terminalia petiolaris* ethyl acetate. Positive controls = ampicillin (AMP) 2 μg and erythromycin (ERTH) 10 μg. Negative control (NC) = water. Results are expressed as the mean zones of inhibition of three replicates ± SEM. * indicates results that are significantly different to the negative control (*p* < 0.01). The horizontal line at 6 mm on the *y*-axis indicates the diameter of the discs used in the assay.

**Figure 2 antibiotics-12-01643-f002:**
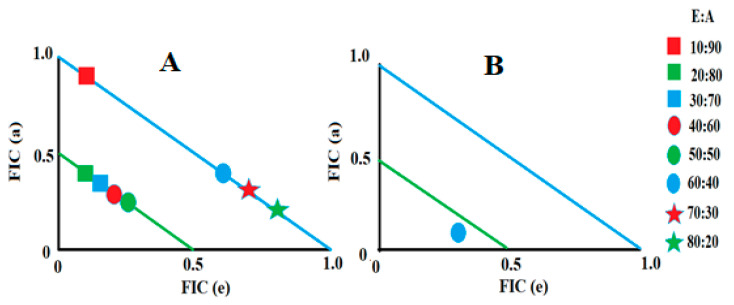
(**A**) Isobologram of the TPW extract in combination with tetracycline when tested at various ratios against *S. aureus*. (**B**) Isobologram of the TPE extract in combination with erythromycin when tested at various ratios against *K. pneumoniae.* Results represent the mean MIC values of two repeats. Ratio = % extract: % antibiotic. Ratios ≤ 0.5/0.5 represent synergy (ΣFIC ≤ 0.5). Any ratio > 0.5/0.5 and ≤1/1 are considered additive (ΣFIC > 0.5–1.0). Only synergistic and additive ratios are displayed.

**Figure 3 antibiotics-12-01643-f003:**
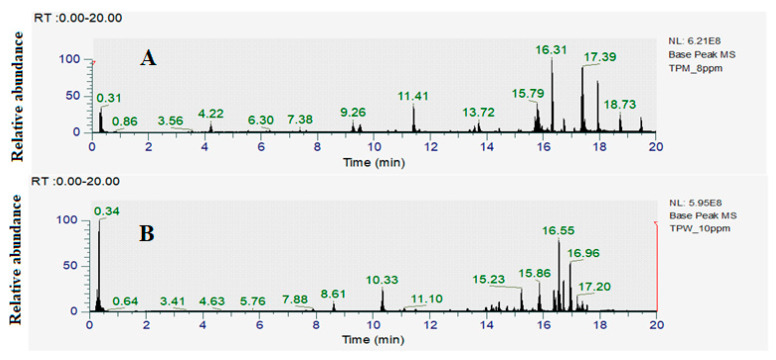
Positive ion LC-MS total compound chromatograms of (**A**) TPM (*Terminalia petiolaris* methanol) and (**B**) TPW (*Terminalia petiolaris* water) extracts.

**Figure 4 antibiotics-12-01643-f004:**
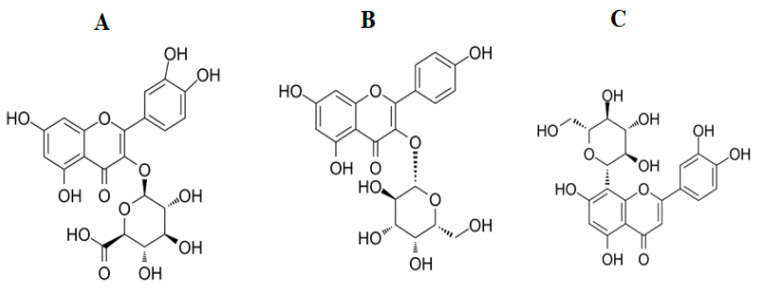
Chemical structure of (**A**) miquelianin, (**B**) trifolin and (**C**) orientin.

**Table 1 antibiotics-12-01643-t001:** MIC values (µg/mL) of plant extract alone and in combination with conventional antibiotics against the bacteria tested in this study.

Extract and Antibiotic	MIC (µg/mL)					
	*E. coli*	ESBL *E. coli*	*S. aureus*	MRSA	*K. pneumoniae*	ESBL *K. pneumoniae*
TPM	109.37	437.5	437.5	437.5	437.5	437.5
TPW	218.75	875	875	875	218.75	212.5
TPE	250	250	250	250	250	250
Tetracycline	-	-	1.25	-	-	-
Chloramphenicol	-	-	0.31	-	1.25	1.25
Ciprofloxacin	2.5	-	0.62	2.5	2.5	1.25
Gentamicin	0.039	0.039	0.03	0.03	0.03	0.03
Erythromycin	-	-	1.25	-	2.5	-
Negative control	-	-	-	-	-	-

MIC values for TPM = *Terminalia petiolaris* methanol, TPW = *Terminalia petiolaris* water, TPE = *Terminalia petiolaris* ethyl acetate and LD = liquid dilution; - indicates no inhibition at any dose observed. MIC values of triplicate determinations are shown and expressed in units of µg/mL.

**Table 2 antibiotics-12-01643-t002:** ∑ FIC values obtained from interactions between plant extracts and antibiotics.

Bacteria	Extract	Tetracycline	Chloramphenicol	Ciprofloxacin	Gentamicin	Erythromycin
*E. coli*	TPM	-	-	0.52	3.60	-
	TPW	-	-	2.25	0.71	
	TPE	-	-	0.75	1.36	-
ESBL *E. coli*	TPM	-	-	-	2.85	-
	TPW	-	-	-	5.33	-
	TPE	-	-	-	2.66	-
*S. aureus*	TPM	0.62	1	1.50	0.71	0.62
	TPW	**0.50**	1.25	1.04	0.66	1
	TPE	1	1.25	1.5	0.66	1
MRSA	TPM	-	-	2.25	2.85	-
	TPW	-	-	0.75	0.66	-
	TPE	-	-	0.75	2.60	-
*K. pneumoniae*	TPM	-	0.62	1.12	2.85	0.56
	TPW	-	0.62	2.25	0.71	1.12
	TPE	-	2	0.75	0.66	**0.40**
ESBL *K. pneumoniae*	TPM	-	0.62	1.25	2.85	-
	TPW	-	0.63	2.55	1.42	-
	TPE	-	2	1	0.66	-

∑ FIC values of plant extracts in combination with conventional antibiotics against sensitive and resistant strains of *E. coli*, *S. aureus* and *K. pneumoniae*. TPM = *Terminalia petiolaris* methanol; TPW = *Terminalia petiolaris* water; TPE = *Terminalia petiolaris* ethyl acetate. **Synergy ≤ 0.5**; 0.5 ≤ *Additive* ≤ 1.0, 1.0 ≤ Indifferent ≤ 4; Antagonistic ≥ 4.0. FIC values were performed in duplicate. - indicates no inhibition at any concentration tested.

**Table 3 antibiotics-12-01643-t003:** Qualitative analysis of LC-MS of TPM and TPW.

Retention Time (Min)	Empirical Formula	Molecular Mass	Putative Identification	Relative Abundance (% Total Area)	
				TPM	TPW
7.06	C_21_H_20_O_11_	448	Trifolin	1.36	
6.35	C_27_H_30_O_16_	610	Quercetin	0.18	
5.54	C_21_H_20_O_11_	448	Orientin	5.17	
5.94	C_21_H_18_O_13_	478	Miquelianin	0.07	
6.87	C_28_H_24_O_15_	600	Isoorientin 2″-*O*-gallate	0.04	
6.37	C_24_H_26_O_9_	458	7-Hydroxy-5,4′-dimethoxy-8-methylisoflavone 7-*O*-rhamnoside	0.05	
7.37	C_23_H_22_O_13_	506	6-Methoxyluteolin 7-glucuronide methyl ester	2.73	
7.16	C_22_H_20_O_13_	492	6-Methoxyluteolin 7-glucuronide	0.12	
0.32	C_20_H_18_O_13_	466	2-(3,4-Dihydroxyphenyl)-3,5,7-trihydroxy-8-{[(2*R*,3*R*,4*S*,5*S*,6*R*)-3,4,5,6-tetrahydroxytetrahydro-2*H*-pyran-2-yl]oxy}-4*H*-chromen-4-one	0.42	
7.97	C_15_H_10_O_7_	302	2-(2,4-Dihydroxyphenyl)-3,5,7-trihydroxy-4*H*-chromen-4-one	0.05	
6.73	C_28_H_24_O_14_	584	2″-*O*-Galloylisovitexin	0.10	
6.18	C_21_H_20_O_10_	432	1,5-Anhydro-1-[5,7-dihydroxy-3-(4-hydroxyphenyl)-4-oxo-4*H*-chromen-8-yl]hexitol	1.30	
6.21	C_28_H_24_O_15_	600	(2*S*,3*R*,4*R*,5*S*,6*S*)-2-{[2-(3,4-Dihydroxyphenyl)-5,7-dihydroxy-4-oxo-4*H*-chromen-3-yl]oxy}-3,5-dihydroxy-6-methyloxan-4-yl 3,4,5-trihydroxybenzoate	0.98	
6.17	C_20_H_22_O_10_	422	(2*R*,3*S*)-7-{[(2*S*,3*R*,4*R*,5*S*)-3,4-Dihydroxy-5-(hydroxymethyl)oxolan-2-yl]oxy}-2-(3,4-dihydroxyphenyl)-3,4-dihydro-2*H*-1-benzopyran-3,5-diol	2.55	
0.52	C_15_H_12_O_7_	322	(2*R*,3*R*)-2-(2,6-Dihydroxyphenyl)-3,5,7-trihydroxy-2,3-dihydro-4*H*-chromen-4-one	0.24	
6.29	C_21_H_20_O_11_	448	Orientin		1.55
6.95	C_28_H_24_O_15_	600	(2*S*,3*R*,4*R*,5*S*,6*S*)-2-{[2-(3,4-Dihydroxyphenyl)-5,7-dihydroxy-4-oxo-4*H*-chromen-3-yl]oxy}-3,5-dihydroxy-6-methyloxan-4-yl 3,4,5-trihydroxybenzoate		0.37
7.36	C_21_H_20_O_10_	432	1,5-Anhydro-1-[5,7-dihydroxy-3-(4-hydroxyphenyl)-4-oxo-4H-chromen-8-yl]hexitol		0.17

## Data Availability

All data are available from the corresponding author on reasonable request.

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
