# Peer review of "Terminalia petiolaris A.Cunn ex Benth. Extracts Have Antibacterial Activity and Potentiate Conventional Antibiotics against β-Lactam-Drug-Resistant Bacteria"

_antibiotics, 2023, doi:10.3390/antibiotics12111643_

Round 1

Reviewer 1 Report

Comments and Suggestions for Authors

Dear authors. 

The manuscript presented original and relevant information about the analyzed topic. However, minimal observation needs to be attended: 

Line 184. Add reference. 

Add a discussion about the toxicity studies. 

Line 301. Add an identification voucher of the analyzed plant. 

Author Response

Responses to Reviewer 1

Manuscript ID: antibiotics-2719280

Terminalia petiolaris A.Cunn ex Benth. Extracts Have Antibacterial Activity and Potentiate Conventional Antibiotics Against β-Lactam-drug Resistant Bacteria (Manuscript ID antibiotics-2719280)

Reviewer 1

Dear authors. The manuscript presented original and relevant information about the analyzed topic. However, minimal observation needs to be attended:

Line 184. Add reference.

A reference was already added to the next sentence regarding the diffusion of large molecules in agar gels. That reference is also valid for the statement about polarity and diffusion. Therefore, [23] has now also been added as a reference for that statement as well for clarity.

Add a discussion about the toxicity studies.

Whilst the toxicity had been discussed in the results, we agree with the reviewer that a greater discussion was required. To address this point, we have added the following paragraph at the end of the discussion section”

“The methanol and ethyl acetate extracts screened in this study were nontoxic in the Artemia nauplli bioassay, indicating that they are safe for antibiotic use. However, further testing in a panel of human cell lines is required before these extracts are used medicinally. Of concern, the aqueous extract displayed toxicity in this assay and therefore it should only be used with caution. However, fractionation of that extract may be able to separate antibacterial compounds from toxic compounds. Further studies are required to explore this possibility.”

Line 301. Add an identification voucher of the analyzed plant.

To address the reviewer’s comment, this has now been revised to “The plant material was supplied as fresh leaves and a voucher specimen is stored in the School of Environment and Science, Griffith University (TP_WAJM21). The leaves were dried.,..”

Reviewer 2 Report

Comments and Suggestions for Authors

The publication submitted for review "Terminalia petiolaris A.Cunn ex Benth. Extracts Have Antibacterial Activity …” concerns the determination of the antibacterial activity of the tested extracts and their possibility of use as potential antibiotics. The publication is interesting, it shows, among others, the fact of the synergistic and additive interactions of the selected combinations of extracts and antibiotics. The authors mentioned why the extracts exhibit different antibacterial activities.

However, I have a few comments:

1) In the line 116 there is erthyromycin.

2) In the table 3, please write in italics the H and O atoms in the names of the compounds, e.g. Isoorientin 2”-O-gallate and similarly with hydrogen atoms in the names in the following lines.

3) For optically active compounds, the type of stereoisomer should be written in italics in the name, e.g. (2R,3R)-2-(2,6-Dihydroxyphenyl)…

4) The reference list contains reference 43 (line 334), the publication contains references 1-36.

Author Response

Responses to Reviewer 2

Manuscript ID: antibiotics-2719280

Terminalia petiolaris A.Cunn ex Benth. Extracts Have Antibacterial Activity and Potentiate Conventional Antibiotics Against β-Lactam-drug Resistant Bacteria (Manuscript ID antibiotics-2719280)

Reviewer 2

The publication submitted for review "Terminalia petiolaris A.Cunn ex Benth. Extracts Have Antibacterial Activity …” concerns the determination of the antibacterial activity of the tested extracts and their possibility of use as potential antibiotics. The publication is interesting, it shows, among others, the fact of the synergistic and additive interactions of the selected combinations of extracts and antibiotics. The authors mentioned why the extracts exhibit different antibacterial activities.

However, I have a few comments:

1) In the line 116 there is erthyromycin.

We thank the reviewer for pointing out this spelling mistake. This has now been revised to “erythromycin”

2) In the table 3, please write in italics the H and O atoms in the names of the compounds, e.g. Isoorientin 2”-O-gallate and similarly with hydrogen atoms in the names in the following lines.

All instances of H and O have now been italicised in Table 3. Additionally, we have also now italicised the R and S.

3) For optically active compounds, the type of stereoisomer should be written in italics in the name, e.g. (2R,3R)-2-(2,6-Dihydroxyphenyl)…

We have also now italicised the R and S in Table 3.

4) The reference list contains reference 43 (line 334), the publication contains references 1-36.

This was an error from a previous version. This has now been revised to [22].

Reviewer 3 Report

Comments and Suggestions for Authors

RESULTS      (Line 85)

2.4 Identification of compounds in TPM and TPW extracts, Line 141

Lines 163 to 165:  A range of different compounds were identified in the TPM and TPW extract of which 14 and 3 were flavonoids, respectively (Table 3).

Question:  In Table 3, where and how to identify compounds 3 and 14? Respectively to whom?

 DISCUSSION      (Line 170)

Lines 191-192: Notably, the T. petiolaris extracts exhibited comparable inhibitory effects on antibiotic resistant bacterial species compared to their susceptible counterparts.

Question: In my understanding, the end of the sentence is not clear: ...... compared to their susceptible counterparts.

Paragraph 1, Line 275: Figure 3: An oxygen atom is missing in the glycosidic part of the C structure

Author Response

Responses to Reviewer 3

Manuscript ID: antibiotics-2719280

Terminalia petiolaris A.Cunn ex Benth. Extracts Have Antibacterial Activity and Potentiate Conventional Antibiotics Against β-Lactam-drug Resistant Bacteria (Manuscript ID antibiotics-2719280)

Reviewer 3

RESULTS (Line 85)

2.4 Identification of compounds in TPM and TPW extracts, Line 141

This has now been revised to “…T. petiolaris methanol (TPM) and aqueous (TPW extracts)”

Lines 163 to 165:  A range of different compounds were identified in the TPM and TPW extract of which 14 and 3 were flavonoids, respectively (Table 3).

Question:  In Table 3, where and how to identify compounds 3 and 14? Respectively to whom?

We thank the reviewer for pointing this out. The following text has been added to the end of section 4.9 to address this point:

“The data were analysed using Compound DiscovererTM 3.2 software with a Fragment Ion Search (FISh) function. The putative compound identification (where possible) and molecular annotation used the Global Natural Product Social Molecular Networking (GNPS), mzCloud, mzVault, CyanoMetDB, and Chemspider databases and were compared to published data.”

DISCUSSION (Line 170)

Lines 191-192: Notably, the T. petiolaris extracts exhibited comparable inhibitory effects on antibiotic resistant bacterial species compared to their susceptible counterparts.

Question: In my understanding, the end of the sentence is not clear: ...... compared to their susceptible counterparts.

This has now been revised to the following to address the reviewer’s comment:

Notably, the T. petiolaris extracts exhibited comparable inhibitory effects on antibiotic-resistant bacterial species compared to the susceptible counterparts of the same bacterial species.”

Paragraph 1, Line 275: Figure 3: An oxygen atom is missing in the glycosidic part of the C structure

We have checked this structure and we believe that it is correct as it is. We understand the reviewer’s belief that this is incorrect as orientin is an unusual structure. The gluco- moiety is not joined by a glycosidic linkage to the flavonoid moiety. Instead, the gluco- carbon is bound directly into a carbon in the flavonoid moiety. Therefore, whilst this structure is unusual, it is correct as shown.